

# A multi-center cross-sectional investigation of *BRAF* V600E mutation in Ameloblastoma

Khin Mya Tun[1], Puangwan Lapthanasupkul[2], Anak Iamaroon[3], Wacharaporn Thosaporn[3], Poramaporn Klanrit[4], Sompid Kintarak[5], Siwaporn Thanasan[6], Natchalee Srimaneekarn[7] and Nakarin Kitkumthorn[8]

[1] Faculty of Dentistry, Mahidol University, Bangkok, Thailand
[2] Department of Oral and Maxillofacial Pathology, Faculty of Dentistry, Mahidol University, Bangkok, Thailand
[3] Department of Oral Biology and Diagnostic Sciences, Faculty of Dentistry, Chiang Mai University, Chiang Mai, Thailand
[4] Department of Oral Biomedical Sciences, Faculty of Dentistry, Khon Kaen University, Khon Kaen, Thailand
[5] Department of Stomatology, Faculty of Dentistry, Prince of Songkla University, Songkhla, Thailand
[6] Department of Pathology, Rajavithi Hospital, Bangkok, Thailand
[7] Department of Anatomy, Faculty of Dentistry, Mahidol University, Bangkok, Thailand
[8] Department of Oral Biology, Faculty of Dentistry, Mahidol University, Bangkok, Thailand

Corresponding author
Nakarin Kitkumthorn,
nakarinkit@gmail.com

## ABSTRACT

**Background**. B-Raf proto-oncogene, serine/threonine kinase (*BRAF*) V600E mutation stands as a pivotal genetic alteration strongly associated with several neoplasms and contributes significantly to their pathogenesis as well as potential targeted treatment strategies.

**Objective**. This cross-sectional study aimed to determine the frequency of *BRAF* V600E mutation in ameloblastoma in a multi-center of Thailand.

**Method**. Anti-BRAF V600E (clone VE1) immunohistochemistry was performed on 227 conventional ameloblastoma (AM) and 113 unicystic ameloblastoma (UA) samples collected from four major dental schools located in the Central, North, South, and Northeast regions of Thailand. Tumor cells from randomly chosen AM cases were also micro-dissected from the FFPE sections and subjected to DNA sequencing to confirm the immunohistochemical results.

**Results**. *BRAF* V600E mutation was detected in 71.8% of the AM samples, while 65.5% of samples with UAs demonstrated *BRAF* V600E positivity. The *BRAF* V600E mutation was significantly different in the histological subtypes of AMs in the four centers ($p = 0.012$) and the location of UA in three centers ($p = 0.013$). There was no significant association between the *BRAF* V600E mutation and the location of ameloblastoma in the overall prevalence of our multi-center study; nonetheless, a statistically significant association was found between the *BRAF* V600E mutation and the mandible location of AMs from the Central Faculty of Dentistry, Mahidol University (MU) center ($p = 0.033$), as well as with the histological subtypes of AMs from the Southern Faculty of Dentistry, Prince of Songkla University (PSU) center ($p = 0.009$). No statistical association was observed between the *BRAF* V600E mutation and AM and UA recurrence ($p = 0.920$ and $p = 0.312$), respectively. The results of DNA sequencing performed in randomly selected 40 *BRAF* V600E-positive and 20 *BRAF* V600E-negative ameloblastoma tissues were in accordance with the immunohistochemical findings.

**Conclusion**. As a result of a notable prevalence of *BRAF* V600E in Thai individuals diagnosed with ameloblastoma, they may benefit from the utilization of adjunctive anti-BRAF targeted therapy for treatment.

# INTRODUCTION

Ameloblastoma is the most common locally invasive benign odontogenic tumor. This slow-growing jaw tumor of odontogenic origin is implicated in approximately 1% of all oral tumors and 9%–11% of all odontogenic tumors (*Masthan et al., 2015*). Although the lesions can develop in the mandible or the maxilla, 80% of ameloblastomas occur in the mandibular molar region (*Petrovic et al., 2018*). In the fifth edition of the World Health Organization (WHO) Classification of Head and Neck Tumors (2022) (*WHO Classification of Tumours Editorial Board, 2022*), ameloblastomas were classified as unicystic ameloblastoma (UA), conventional ameloblastoma (hereafter referred to as AM), extraosseous/peripheral ameloblastoma and metastasizing ameloblastoma. In addition, adenoid ameloblastoma was also added as a new distinct entity in the group of benign epithelial odontogenic tumors (*WHO Classification of Tumours Editorial Board, 2022*).

Most cases of AM demonstrate aggressive behaviors that can potentially invade tissues, resorb the nearby tooth roots, infiltrate medullary spaces, and erode cortical bone (*Kumar et al., 2021*; *Qiao et al., 2021*). Wide surgical excision is typically considered as the most effective treatment for AM. However, it may lead to significant facial disfigurement and substantial morbidity. The recurrence rate of ameloblastoma varies substantially ranging from 55% to 90% (*Bera & Tiwari, 2024*). Hence, efforts are underway to reduce the invasiveness of these treatments. Rapid advancements in gene mutation research and the application of targeted drug therapy to combat cancer offer less aggressive alternatives to surgical excision.

B-Raf proto-oncogene, serine/threonine kinase (*BRAF*) is frequently affected by a somatic point mutation of *BRAF* V600E in human cancers. The valine (V) to glutamic acid (E) substitution at codon 600 (*BRAF* V600E) represents a *BRAF* missense mutation that constitutively activates the mitogen-activated protein kinase (MAPK) pathway (*Michaloglou et al., 2008*). The *BRAF* V600E mutation has been identified as the most common oncogenic driver mutation in malignant melanoma, thyroid cancer, colon cancer, and other malignant neoplasms (*Ritterhouse & Barletta, 2015*).

In our previous study conducted in 2020, a substantial occurrence of this mutation was observed in a cohort of Thai patients with AM and UA in the central region of Thailand (*Lapthanasupkul et al., 2021*). However, the information on *BRAF* V600E mutation frequency in AM and UA in other regions of Thai population is still scarce. Therefore, this study aimed to determine the prevalence of this mutation in AM and UA patients at four major dental schools in different regions (Central, North, South, and Northeast) of

Thailand. Additionally, we conducted a clinicopathological association analysis to identify any clinicopathological factors associated with the *BRAF* V600E mutation.

## MATERIALS & METHODS

### Study design and tissue samples

A cross-sectional design was used to investigate the frequencies of *BRAF* V600E mutation in AM and UA tissue samples; collected from four major dental schools in Thailand's central, north, south, and northeast regions. This study was approved by the Institutional Review Board of the Faculty of Dentistry/Faculty of Pharmacy, Mahidol University (COA.No.MU-DT/PY-IRB 2020/032.0906), Human Experimentation Committee of the Faculty of Dentistry, Chiang Mai University (No. 45/2020), Human Research Ethics Committee of the Faculty of Dentistry, Prince of Songkla University (EC6308-032), and the Ethics Committee of the Human Research Panel 2 of the Faculty of Medicine, Khon Kaen University. A total of 340 formalin-fixed paraffin-embedded (FFPE) samples were collected from the four centers. The delegate centers consisted of the following: the Faculty of Dentistry, Mahidol University (MU), in Bangkok province, representing the Central region; the Faculty of Dentistry, Chiang Mai University (CMU), in Chiang Mai province, representing the Northern region; the Faculty of Dentistry, Khon Kaen University (KKU), in Khon Kaen province, representing the Northeastern region; and the Faculty of Dentistry, Prince of Songkla University (PSU) in Songkla province, representing the Southern region. A total of 227 AM samples from MU, CMU, KKU, and PSU, 113 UA samples from MU, CMU, and PSU were included in this study. All experiments were performed at the laboratory of Oral and Maxillofacial Pathology at Mahidol University. The clinical information about the patients (age and sex) and the tumors, duration (time since the patient first noticed the tumor), tumor size, location, and recurrence were retrieved from the pathology request forms or clinical chart records. The exclusion criteria are low amounts of pathologic tissues and a lack of clinical data. Written informed consent was obtained from all patients. The histological variants of AM are categorized into follicular, plexiform, and mixed type (a combination of follicular and plexiform types). The histological variants of UA are classified as luminal, intraluminal, and mural in this study. The histological diagnosis of all of the AM and UA subtypes was confirmed by two pathologists (KMT and PL).

### VE1 immunohistochemistry

*BRAF* V600E mutation was observed using VE1 immunohistochemical staining, as described previously (*Lapthanasupkul et al., 2021*). Briefly, the FFPE samples were cut into 3-μm-thick sections using a microtome (Microm HM355S, Thermo Fisher Scientific, Walldorf, Germany). Immunohistochemistry with anti-BRAF V600E VE1 mouse monoclonal primary antibody (catalog number 760-5095, Ventana Medical Systems, Tucson, AZ, US) was performed using an automated Ventana BenchMark Ultra autostainer (Ventana Medical Systems, Tucson, AZ, US) with incubation at 37 °C, 1 h. VE1 immunoreactivity was visualized using an OptiView DAB IHC detection kit (Ventana Medical Systems, Tucson, AZ, US) and then counterstained with Hematoxylin

II and Bluing Reagent for 16 min and 4 min, respectively. Melanoma tissues known to be positive and negative for *BRAF* V600E mutation were used as positive control and negative control, respectively (*Meevassana et al., 2022*). Both control tissues were included in each run of the experiment. Two pathologists (KMT and PL), who scored the *BRAF* V600E status, were blinded to the clinicopathological data at the time of interpretation. Immunoreactions were evaluated as positive, when cytoplasmic staining was observed for a significant number of tumor cells (>80%) in the sections. A negative score was recorded for weak nuclear staining, isolated nuclear staining, weak staining of single interspersed cells, and staining of monocytes/macrophages (*Lapthanasupkul et al., 2021*).

## Tissue microdissection and DNA extraction

*BRAF* V600E immunohistochemical 40 positive cases and 20 negative cases of AM were randomly selected for manual microdissection, which was performed as described previously (*Kitkumthorn & Mutirangura, 2010*). Briefly, 5-μm-thick sections of FFPE blocks were serially cut into five levels. The first and last levels were stained with Hematoxylin and Eosin (H&E) to ensure that the outlines of the samples across all slides were identical. The area under the microscope was examined and outlined with a Startmark-pen™ (NC1523755, Thermo Fisher Scientific, Waltham, MA, US). Using the H&E slides as a reference, the tumor areas were marked on the remaining unstained slides (levels 2–4). Finally, the selected areas were dissected using a sterile needle-gauge 21 and stored in phosphate-buffered saline until DNA extraction.

Genomic DNA was extracted using the standard phenol-chloroform protocol (*Sambrook & Russell, 2006*), and the concentration was evaluated using a NanoDrop 2000 spectrophotometer (ND-1000 Spectrophotometer, NanoDrop Technologies, Wilmington, DE, US). A 260/280 optical density ratio of 1.8 was considered acceptable for DNA purity.

## *BRAF* V600E DNA sequencing

DNA sequencing was conducted to validate the *BRAF* V600E immunostaining results. DNA from micro-dissected tumor tissues was amplified and sequenced at *BRAF* exon 15 using the following primers: forward 5′-GAAATTAGATCTCTTACCTAAACTCTTCATA-3′ and reverse 5′-GACCCACTCCATCGAGATTT-3′. The results were compared with the *BRAF* nucleotide sequence available in the National Center for Biotechnology Information database using the MEGA X (Molecular Evolutionary Genetics Analysis) software, version 10.2.4 (MEGA-X, Allenstown, NH, US) to detect the mutation sequence (*Kumar et al., 2018*).

## Statistical analysis

The frequency of *BRAF* V600E mutation was reported as a percentage. Chi-square test was used to investigate the associations between clinicopathologic variables and *BRAF* V600E status in all samples and subgroup analyses. A *p*-value of less than 0.05 was regarded as statistically significant. SPSS, version 28.0 (IBM, Armonk, NY, USA) was used to analyze the data in this study.

## RESULTS

### Clinicopathological characteristics and *BRAF* V600E mutation in AMs

As shown in Table 1, 227 AM patients (125 males and 102 females) were included in the study. The median age of the patients was 33 years (range, 11–79 years). The tumors were located in the mandible in the majority of cases (90%), and the median tumor size was four cm (range, 0.8–15 cm). The average duration of follow-up was 6 months (range, 3.5–30 months). Multilocular AM was more frequent than unilocular AM (66% *vs.* 34%). Furthermore, 40% of the AM cases were histologically diagnosed as follicular, followed by plexiform (28%) and mixed (32%) types. The *BRAF* V600E mutation significantly differed in the histological subtypes of AM in the four centers ($p = 0.012$, Table 1).

BRAF V600E staining was positive in 71.8% of the total AM cases from the four centers (Table 1). The negative cases lacked the cytoplasmic expression of *BRAF* V600E protein in the ameloblastic epithelium, thus indicating the expression of wild-type *BRAF* (Fig. 1A). The positive cases revealed the brown-colored cytoplasmic staining of *BRAF* V600E protein in the ameloblastic epithelium, whereas no staining was observed in the stromal components and non-neoplastic tissues (Fig. 1B). In concordance with the immunostaining results, *BRAF* sequencing revealed a base substitution (GTG > GAG) in all 40 cases with positive *BRAF* V600E immunostaining (Fig. 1D). In contrast, no base substitution was detected in the 20 cases, being negative for *BRAF* V600E (Fig. 1C).

As demonstrated in Table 2, the number of patients with *BRAF* V600E mutation was slightly different at each center: 75.9% at the Central MU center, 68.5% at the Northern CMU center, 61.4% in the Southern PSU center, and 84.9% in the Northeastern KKU center. A statistically significant ($p < 0.05$) association was observed between the presence of *BRAF* V600E mutation and the mandible location of the AMs from the Central MU center ($p = 0.033$) and the histological subtypes of the AMs from the Southern PSU center ($p = 0.009$) (Table 2). Detailed data on AM is presented in Tables S1–S4.

### Clinicopathological characteristics and *BRAF* V600E mutation in UA

As shown in Table 3, 113 patients (median age, 20 years; range, 1–77 years) diagnosed with UA from the Central MU, Northern CMU and Southern PSU centers were enrolled in this study. The median tumor size was three cm (range, 0.9–10 cm), and most lesions occurred in the mandible (89.4%). The average follow-up duration was 6 months (range, 6–26.5 months). Regarding the histological subtype, the mural subtype accounted for 65.5% of the cases, followed by the intraluminal (21.2%), and luminal (13.3%) subtypes in Table 3.

The absence of cytoplasmic *BRAF* V600E protein expression in the cystic epithelium of UA is indicative of wild-type *BRAF* (Fig. 1E). In contrast, the cytoplasmic *BRAF* V600E expression observed in the UA cystic epithelial lining cells indicates the presence of a *BRAF* V600E mutation in UA (Fig. 1F). Overall, 65.5% of UAs were positive for *BRAF* V600E (Table 3). A statistically significant association was only observed between the presence of *BRAF* V600E mutation and the location of the mandible ($p = 0.013$) in three dental centers (Table 3) and ($p = 0.004$) for mandibular cases at the MU center (Table 4). Interestingly, UA patients from the Central MU center (92.3%) had a considerably higher rate of *BRAF*

**Table 1  Clinicopathological characteristics of *BRAF* V600E mutation in conventional ameloblastomas from four dental centers.**

| Clinical characteristics | *BRAF* wild type N (row %) 64 (28.2) | *BRAF* V600E N (row%) 163 (71.8) | Total number N (%) 227 (100) | *p*-value |
|---|---|---|---|---|
| Age (years; Median 33) | | | | |
| ≤33 years | 36 (30.5) | 82 (69.5) | 118 (52) | 0.420 |
| >33 years | 28 (25.7) | 81 (74.3) | 109 (48) | |
| Sex | | | | |
| Male | 38 (30.4) | 87 (69.6) | 125 (55.1) | 0.413 |
| Female | 26 (25.5) | 76 (74.5) | 102 (44.9) | |
| Tumor duration | | | | |
| ≤6 months | 27 (28.7) | 67 (71.3) | 94 (41.4) | |
| >6 months- 1 year | 18 (31.6) | 39 (68.4) | 57 (25.1) | 0.698 |
| ≥1 year | 19 (25) | 57 (75) | 76 (33.5) | |
| Size (Median 4 cm) | | | | |
| ≤4 cm | 31 (26.7) | 85 (73.7) | 116 (51.1) | 0.615 |
| >4 cm | 33 (29.7) | 78 (70.3) | 111 (48.9) | |
| Location | | | | |
| Maxilla | 9 (39.1) | 14 (60.9) | 23 (10.2) | 0.219 |
| Mandible | 55 (27.1) | 149 (73) | 204 (89.8) | |
| Radiographic Feature | | | | |
| Unilocular | 23 (29.9) | 54 (70.1) | 77 (33.9) | 0.688 |
| Multilocular | 41 (27.3) | 109 (72.7) | 150 (66.1) | |
| Histologic Subtype | | | | |
| Follicular | 22 (24.2) | 69 (75.8) | 91 (40.1) | |
| Plexiform | 27 (42.2) | 37 (57.8) | 64 (28.2) | 0.012[*] |
| Mixed | 15 (20.8) | 57 (79.2) | 72 (31.7) | |
| Recurrence | | | | |
| Absent | 58 (28.3) | 147 (71.7) | 205 (90.3) | 0.920 |
| Present | 6 (27.3) | 16 (72.7) | 22 (9.7) | |

**Notes.**
[*]$p < 0.05$ (Chi-square test).

V600E mutation than those from the Northern CMU (43.8%, $p < 0.001$) and the Southern PSU centers (41.4%, $p < 0.001$; Table 4). Further details are presented in Tables S5–S7.

## DISCUSSION

Ameloblastoma is characterized by excessive cell proliferation, which is primarily regulated by the mitogen-activated protein kinase (MAPK) signaling pathway (*Brown & Betz, 2015*). Mutations in the MAPK pathway have been shown to play a key role in the pathogenesis of AM, especially the *BRAF* V600E (*Brown & Betz, 2015*; *Kurppa et al., 2014*). *BRAF* V600E immunostaining was positive in 71.8% of the total AM cases from the four dental centers in Thailand. Our result is consistent with the previous meta-analysis reporting a pooled prevalence of 70.49% for *BRAF* V600E in 833 AM cases (*Mamat Yusof, Ch'ng & Radhiah Abdul Rahman, 2022*). Although varying frequencies (33.3%–92%)

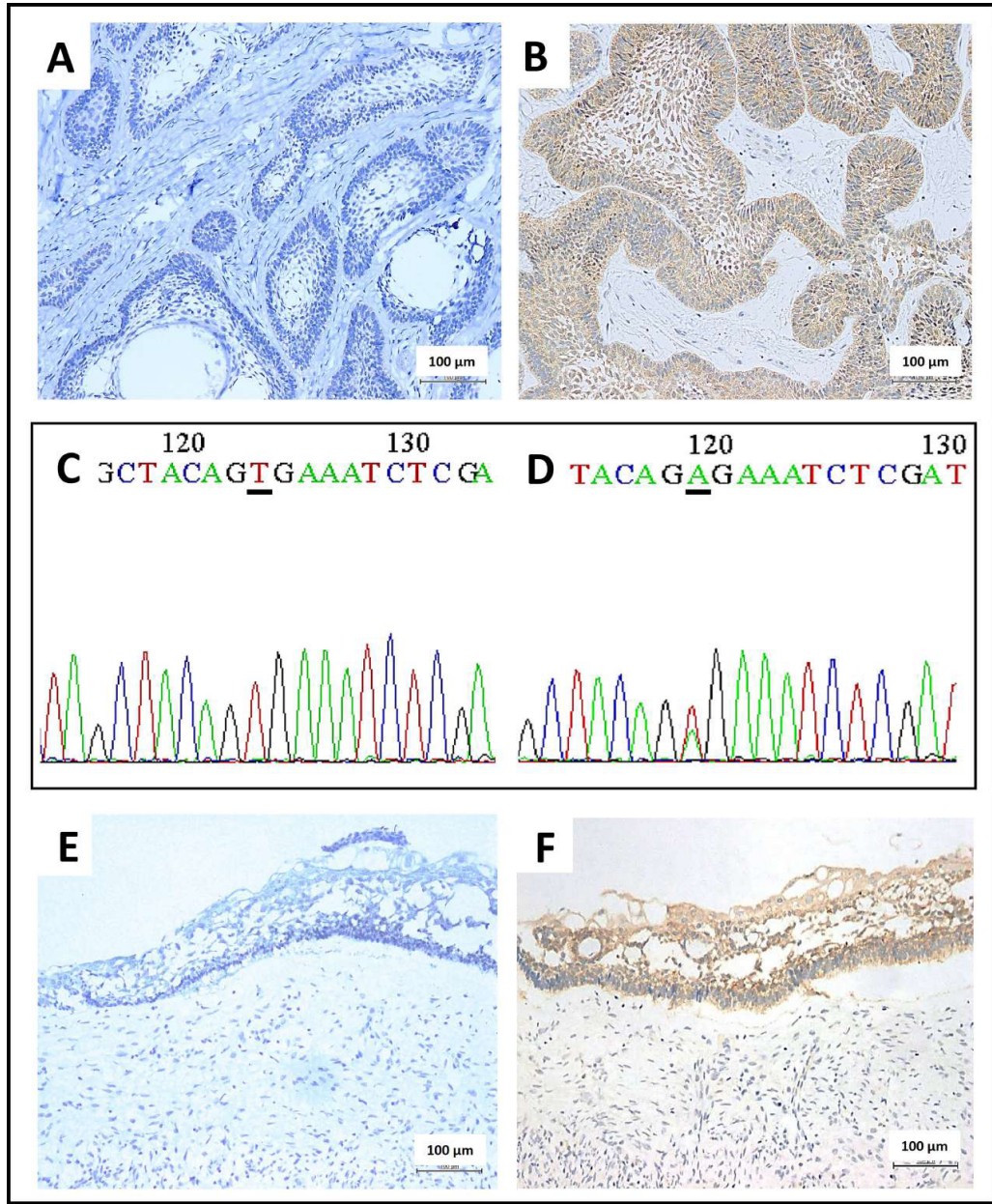

**Figure 1** **Immunohistochemical staining and Sanger sequencing of *BRAF* V600E in ameloblastoma.**
(A) A representative picture of conventional ameloblastoma showed no BRAF V600E protein expression in the odontogenic epithelium (200x magnification). (B) A representative picture of conventional ameloblastoma showed moderate brown cytoplasmic BRAF V600E protein expression in the odontogenic epithelium (200x magnification). (C) DNA sequencing electropherogram of the wild-type allele of *BRAF* V600E mutation (the same case from A), characterized by the nucleotide sequence coding for the amino acid valine in the underline area (A-adenine, C-cytosine, G-guanine, T-thymine) (D) DNA sequencing electropherogram of ameloblastoma, 

**Figure 1 (…continued)**
forward mutation of *BRAF* codon encoding Val600Glu (V600E) (the same case from B.), characterized by the substitution of thymine with adenine in the position of 1799 T > A as shown in the underline region. (E) A representative of unicystic ameloblastoma showed an absence of BRAF V600E protein expression in the cystic lining epithelium (200x magnification). (F) A representative of unicystic ameloblastoma with presence of BRAF V600E protein expression, indicated by brown cytoplasmic staining in the cystic lining epithelium (200x magnification).

of *BRAF* V600E mutation have been documented (*Kokubun et al., 2022*; *Kurppa et al., 2014*; *Lapthanasupkul et al., 2021*; *Shirsat et al., 2018*), the high frequency of *BRAF* V600E mutation in AM remains observed from each center of Thailand, with a range between 61.4% to 84.8%. These findings underscore a crucial role of *BRAF* V600E in the molecular pathogenesis of this common odontogenic tumor, and substantiate a novel alternative therapeutic approach for Thai patients diagnosed with AM.

DNA sequencing was further performed to confirm the immunohistochemical findings in the current study. Similar to the study by *Kokubun et al. (2022)*, *Mendez et al. (2022)*, a high agreement was observed between the immunohistochemical staining and sequencing of *BRAF* V600E in the present study. The superior results observed in the current study may be attributed to meticulous microdissection, enabling precise analysis of the epithelial tumor tissue. However, whereas our analysis included a large number of AM specimens, only a portion of the specimens with tissue available was permitted for DNA sequencing. In addition, the concordance between *BRAF* immunohistochemistry and sequencing techniques has been documented previously in various other tumor types, including xanthoastrocytoma, colorectal cancer, and papillary thyroid cancer (*Day et al., 2015*; *Ida et al., 2013*; *Loo et al., 2018*; *Qiu et al., 2015*).

We found no significant association between *BRAF* V600E mutation and tumor size in any of the AM cases in this study; on the contrary, *Do Canto et al. (2019)* reported a higher frequency of *BRAF* V600E mutant in patients with large AM. There was no significant association between *BRAF* V600E mutation and the location of ameloblastoma in the overall prevalence of our multi-center study. Nevertheless, tumor location at the mandible was significantly associated with *BRAF* V600E mutations only in AM patients from the Central dental center of Thailand. This result is comparable to those reported by *Togni et al. (2022)*, *Bonacina et al. (2022)*, wherein a higher frequency of mandibular AM was associated with *BRAF* V600E mutation. This significant finding may be attributed to different demographic patterns in the Central region, compared to the other centers in Thailand. Therefore, further investigations may be required to validate our findings. Furthermore, in the study of *Kokubun et al. (2022)* no association was observed between *BRAF* V600E mutation and AM recurrence. Although no statistical association was found between *BRAF* mutation and ameloblastoma recurrence, adjunctive anti-BRAF therapy may improve the quality of life. The sequelae of extensive surgical resection, including functional impairment and aesthetic deformities, underscore the need for alternative treatment strategies. BRAF inhibitors may be utilized as an adjuvant presurgical treatment, especially in older patients with large lesions, to assist in conservative surgical interventions for tumor shrinkage and induction of appropriate peripheral bone formation. Such

**Table 2  Clinicopathological characteristics of *BRAF* V600E mutation in conventional ameloblastomas from the individual center.**

| Variables | MU[a] | | CMU[b] | | PSU[c] | | KKU[d] | | p-value |
|---|---|---|---|---|---|---|---|---|---|
| | Wild-type | Mutant | Wild-type | Mutant | Wild-type | Mutant | Wild-type | Mutant | |
| **Number of patients** | | | | | | | | | |
| | 20 (24) | 63 (75.9) | 17 (31.4) | 37 (68.5) | 22 (38.5) | 35 (61.4) | 5 (15.1) | 28 (84.8) | |
| **Age (years; Median 33)** | | | | | | | | | |
| ≤33 years | 9 (18.8) | 39 (81.2) | 11 (42.3) | 15 (57.7) | 14 (50) | 14 (50.0) | 2 (12.5) | 14 (87.5) | 0.182[a] |
| >33 years | 11 (31.4) | 24 (68.6) | 6 (21.4) | 22 (78.6) | 8 (27.6) | 21 (72.4) | 3 (17.6) | 14 (82.4) | 0.099[b] |
| | | | | | | | | | 0.082[c] |
| | | | | | | | | | 0.680[d] |
| **Sex** | | | | | | | | | |
| Male | 13 (30.2) | 30 (69.8) | 13 (38.2) | 21 (61.8) | 10 (32.3) | 21 (67.7) | 2 (11.8) | 15 (88.2) | 0.175[a] |
| Female | 7 (17.5) | 33 (82.5) | 4 (20.0) | 16 (80.0) | 12 (46.2) | 14 (53.8) | 3 (18.8) | 13 (81.2) | 0.164[b] |
| | | | | | | | | | 0.283[c] |
| | | | | | | | | | 0.576[d] |
| **Tumor duration** | | | | | | | | | |
| ≤6 months | 8 (32.0) | 17 (68.0) | 9 (32.1) | 19 (67.9) | 10 (35.7) | 18 (64.3) | 0 | 13 (100) | |
| >6 months–1 year | 7 (25.0) | 21 (75.0) | 6 (42.9) | 8 (57.1) | 2 (33.3) | 4 (66.7) | 3 (33.3) | 6 (66.7) | 0.412[a] |
| ≥1 year | 5 (16.7) | 25 (83.3) | 2 (16.7) | 10 (83.3) | 10 (43.5) | 13 (56.5) | 2 (18.2) | 9 (81.8) | 0.356[b] |
| | | | | | | | | | 0.819[c] |
| | | | | | | | | | 0.095[d] |
| **Location** | | | | | | | | | |
| Maxilla | 4 (57.1) | 3 (42.9) | 1 (50.0) | 1 (50.0) | 3 (42.9) | 4 (57.1) | 1 (14.3) | 6 (85.7) | 0.033[a*] |
| Mandible | 16 (21.1) | 60 (78.9) | 16 (30.8) | 36 (69.2) | 19 (38.0) | 31 (62.0) | 4 (15.4) | 22 (84.6) | 0.566[b] |
| | | | | | | | | | 0.805[c] |
| | | | | | | | | | 0.943[d] |
| **Size (Median 4 cm)** | | | | | | | | | |
| ≤4 cm | 8 (21.6) | 29 (78.4) | 9 (30.0) | 21 (70.0) | 10 (32.3) | 21 (67.7) | 4 (22.2) | 14 (77.8) | 0.636[a] |
| >4 cm | 12 (26.1) | 34 (73.9) | 8 (33.3) | 16 (66.7) | 12 (46.2) | 14 (53.8) | 1 (6.7) | 14 (93.3) | 0.793[b] |
| | | | | | | | | | 0.283[c] |
| | | | | | | | | | 0.215[d] |
| **Radiographic feature** | | | | | | | | | |
| Unilocular | 7 (30.4) | 16 (69.6) | 7 (26.9) | 19 (73.1) | 8 (40.0) | 12 (60.0) | 1 (12.5) | 7 (87.5) | 0.403[a] |
| Multilocular | 13 (21.7) | 47 (78.3) | 10 (35.7) | 18 (64.3) | 14 (37.8) | 23 (62.2) | 4 (16.0) | 21 (84.0) | 0.785[b] |
| | | | | | | | | | 0.873[c] |
| | | | | | | | | | 0.810[d] |
| **Histologic subtype** | | | | | | | | | |
| Follicular | 6 (25.0) | 18 (75.0) | 2 (11.8) | 15 (88.2) | 10 (34.5) | 19 (65.5) | 4 (19.0) | 17 (81.0) | 0.529[a] |
| Plexiform | 8 (30.8) | 18 (69.2) | 11 (45.8) | 13 (54.2) | 8 (80.0) | 2 (20.0) | 0 | 4 (100) | 0.690[b] |
| Mixed | 6 (18.2) | 27 (81.8) | 4 (30.8) | 9 (69.2) | 4 (22.2) | 14 (77.8) | 1 (12.5) | 7 (87.5) | 0.009[c*] |
| | | | | | | | | | 0.605[d] |
| **Recurrence** | | | | | | | | | |
| No recur | 17 (22.1) | 60 (77.9) | 15 (32.6) | 31 (67.4) | 21 (41.2) | 30 (58.8) | 5 (16.1) | 26 (83.9) | 0.123[a] |
| Recur | 3 (50.0) | 3 (50.0) | 2 (25.0) | 6 (75.0) | 1 (16.7) | 5 (83.3) | 0 | 2 (100) | 0.669[b] |
| | | | | | | | | | 0.243[c] |
| | | | | | | | | | 0.538[d] |

**Notes.**

MU, Mahidol University; CMU, Chiang Mai University; PSU, Prince of Songkhla University and KKU, Khon Kaen University.

Percentages are given in parentheses.

*$p < 0.05$ (Chi-square test).

**Table 3 Clinicopathological characteristics of *BRAF* V600E mutation in unicystic ameloblastomas from three centers.**

| Clinical characteristics | *BRAF* wild type N (row %) | *BRAF* V600E N (row %) | Total number N (%) | *p*-value |
|---|---|---|---|---|
| | **39 (34.5)** | **74 (65.5)** | **113 (100)** | |
| **Age (years; Median 20)** | | | | |
| ≤20 years | 23 (39) | 36 (61) | 59 (52.2) | 0.296 |
| >20 years | 16 (29.6) | 38 (70.4) | 54 (47.8) | |
| **Sex** | | | | |
| Male | 17 (32.1) | 36 (67.9) | 53 (46.9) | 0.608 |
| Female | 22 (36.7) | 38 (63.3) | 60 (53.1) | |
| **Tumor duration** | | | | |
| ≤6 months | 29 (42.6) | 39 (57.4) | 68 (60.2) | |
| >6 months- 1year | 5 (29.4) | 12 (70.6) | 17 (15) | 0.060 |
| ≥1 year | 5 (17.9) | 23 (82.1) | 28 (24.8) | |
| **Size (Median 3 cm)** | | | | |
| ≤3 cm | 23 (31.9) | 49 (68.1) | 72 (63.7) | 0.447 |
| >3 cm | 16 (39) | 25 (61) | 41 (36.3) | |
| **Location** | | | | |
| Maxilla | 8 (66.7) | 4 (33.3) | 12 (10.6) | 0.013[*] |
| Mandible | 31 (30.7) | 70 (69.3) | 101 (89.4) | |
| **Histologic subtype** | | | | |
| Luminal | 7 (46.7) | 8 (53.3) | 15 (13.3) | |
| Intraluminal | 8 (33.3) | 16 (66.7) | 24 (21.2) | 0.566 |
| Mural | 24 (34.4) | 50 (67.6) | 74 (65.5) | |
| **Recurrence** | | | | |
| Absent | 37 (35.9) | 66 (64.1) | 103 (91.2) | 0.312 |
| Present | 2 (20) | 8 (80) | 10 (8.8) | |

**Notes.**
[*]*p* < 0.05 (Chi-square test).

utilization ensures safer resection with less deformity and better preservation of function (*Grynberg et al., 2024*). Furthermore, the use of BRAF inhibitors as adjuvant presurgical therapy is particularly recommended in younger patients to mitigate potentially detrimental effects on mandibular growth and facial aesthetics (*Grynberg et al., 2024*).

This study revealed a notable disparity of *BRAF* V600E frequency in UA patients from various regions of Thailand. Interestingly, UA from the Central center showed the highest frequency (92.3%) compared to other centers. This unexpected outcome may be associated with differences in demographic data and a slightly higher proportion of the UA samples in the central center. The high frequency of *BRAF* V600E mutation in UA (88.9–100%) was previously reported by *Kelppe et al. (2019)*, *Goes et al. (2023)* and *Heikinheimo et al. (2019)*. These findings implicated that UA may be mainly driven by *BRAF* V600E mutation compared to AM, and supported BRAF inhibitors as a promising future treatment option for UA. UA is frequently found in young patients when the jaws are still growing. The radical treatment of extensive or multiple recurrent UA cases often poses a challenge

**Table 4  Clinicopathological characteristics of *BRAF* V600E mutation in UA cases from the individual center.**

| Variables | MU-UA[a] | | CMU-UA[b] | | PSU-UA[c] | | *p*-value |
|---|---|---|---|---|---|---|---|
| | **Wild-type** | **Mutant** | **Wild-type** | **Mutant** | **Wild-type** | **Mutant** | |
| **Number of the patients** | 4 (7.7) | 48 (92.3) | 18 (56.2) | 14 (43.8) | 17 (58.6) | 12 (41.4) | |
| **Age (years; Median 20)** | | | | | | | |
| ≤20 years | 2 (9.5) | 19 (90.5) | 11 (55.0) | 9 (45.0) | 10 (55.6) | 8 (44.4) | 0.683[a] |
| >20 years | 2 (6.5) | 29 (93.5) | 7 (58.3) | 5 (41.7) | 7 (63.6) | 4 (36.4) | 0.854[b] 0.668[c] |
| **Sex** | | | | | | | |
| Male | 2 (7.1) | 26 (92.9) | 7 (63.6) | 4 (36.4) | 8 (57.1) | 6 (42.9) | 0.872[a] |
| Female | 2 (8.3) | 22 (91.7) | 11 (52.4) | 10 (47.6) | 9 (60.0) | 6 (40.0) | 0.542[b] 0.876[c] |
| **Tumor duration** | | | | | | | |
| ≤6 months | 3 (13.0) | 20 (87.0) | 13 (54.2) | 11 (45.8) | 13 (61.9) | 8 (38.1) | 0.293[a] |
| >6 months- 1year | 1 (9.1) | 10 (90.9) | 3 (60.0) | 2 (40.0) | 1 (100) | 0 | 0.903[b] |
| ≥1 year | 0 | 18 (100) | 2 (66.7) | 1 (33.3) | 3 (42.9) | 4 (57.1) | 0.469[c] |
| **Location** | | | | | | | |
| Maxilla | 2 (40.0) | 3 (60.0) | 3 (75.0) | 1 (25.0) | 3 (100) | 0 | 0.004[a]* |
| Mandible | 2 (4.3) | 45 (95.7) | 15 (53.6) | 13 (46.4) | 14 (53.8) | 12 (46.2) | 0.419[b] 0.124[c] |
| **Size (median 3 cm)** | | | | | | | |
| ≤3 cm | 3 (9.7) | 28 (90.3) | 12 (50.0) | 12 (50.0) | 8 (47.1) | 9 (52.9) | 0.514[a] |
| >3 cm | 1 (4.8) | 20 (95.2) | 6 (75.0) | 2 (25.0) | 9 (75.0) | 3 (25.0) | 0.217[b] 0.132[c] |
| **Histologic Subtype** | | | | | | | |
| Luminal | 0 | 6 (100) | 6 (75.0) | 2 (25.0) | 1 (100) | 0 | 0.707[a] |
| Intraluminal | 1 (6.7) | 14 (93.3) | 3 (60.0) | 2 (40.0) | 4 (100) | 0 | 0.411[b] |
| Mural | 3 (9.7) | 28 (90.3) | 9 (47.4) | 10 (52.6) | 12 (50.0) | 12 (50.0) | 0.119[c] |
| **Recurrence** | | | | | | | |
| No recur | 4 (9.1) | 40 (90.9) | 17 (54.8) | 14 (45.2) | 16 (57.1) | 12 (42.9) | 0.375[a] |
| Recur | 0 | 8 (100) | 1 (100) | 0 | 1 (100) | 0 | 0.370[b] 0.393[c] |

**Notes.**
MU, Mahidol University; CMU, Chiang Mai University; PSU, Prince of Songkhla University; UA, Unicystic ameloblastoma. Percentages are given in parentheses.
*$p < 0.05$ (Chi-square test).

(*Scariot et al., 2012*). Adjunctive treatment with targeted drug therapy may offer a superior clinical outcome for these patients. In addition, consistent with the study by *Heikinheimo et al. (2019)*, a statistically significant association between mandibular UA and *BRAF* V600E was observed in this study ($p = 0.013$), highlighting the probable role of the MAPK pathway in the pathogenesis of UA in the mandible.

Development of targeted therapy for AM is tempting, based on the high frequency of *BRAF* V600E in this disease. A prior clinical investigation focused on the MAPK pathway as a potential therapeutic approach for AM-targeted drug treatment has been documented previously, wherein advanced AM was managed with dabrafenib, a BRAF inhibitor (*Faden & Algazi, 2017*). The treatment resulted in a remarkable shrinkage of the tumor after 8

months. However, it is important to note that this was reported in a single case. BRAF and mitogen-activated extracellular signal-regulated kinase (MEK) inhibitors, including vemurafenib, dabrafenib, and trametinib, have shown significant reductions in tumor size with promising results in AM patients with *BRAF* V600E (*Fernandes et al., 2018*; *Tan et al., 2016*). *BRAF* V600E inhibitors may be used as pre-operative therapy in *BRAF* V600E-positive AM patients to reduce the primary tumor size. However, additional studies with different clinical settings are needed to confirm the outcome of the targeted treatment.

Our study possesses a strength in terms of its scale, representing the largest investigation of AM-*BRAF* V600E mutation conducted in Thailand. This comprehensive investigation covers all four regions of the country. However, a notable limitation of our study is the potential benefit of expanding the sample size to include data from different countries, which could enhance the robustness and generalizability of our findings.

## CONCLUSIONS

Our study revealed a significantly high prevalence of the *BRAF* V600E mutation among patients diagnosed with AM at four dental centers in Thailand. Additionally, a statistically significant association was found between the *BRAF* V600E mutation and the histological subtype of AM, as well as the location of UA in three dental centers, and with the location of AM and UA cases at the MU center. In contrast, other clinicopathological factors, including age, sex, tumor duration, radiological features, and recurrence, showed no significance among ameloblastoma cases with the *BRAF* V600E mutation. Despite the lack of a statistical association between *BRAF* mutation and ameloblastoma recurrence, adjunctive anti-BRAF targeted therapy may improve the quality of life, addressing functional and aesthetic impairments, especially for ameloblastoma patients who cannot undergo extensive surgical intervention. The significant prevalence of the *BRAF* V600E mutation in the pathogenesis of ameloblastoma in Thai patients suggests the future potential use of BRAF inhibitors as targeted therapy, combined with surgical treatment, to improve treatment outcomes.

### Funding
The present study was supported by grants from the International Dental Collaboration of the Mekong River Region (IDCMR) scholarship, Faculty of Dentistry, Mahidol University. The funders had no role in study design, data collection and analysis, decision to publish, or preparation of the manuscript.

### Grant Disclosures
The following grant information was disclosed by the authors:
International Dental Collaboration of the Mekong River Region (IDCMR).
Faculty of Dentistry, Mahidol University.

### Competing Interests
The authors declare there are no competing interests.

## Author Contributions

- Khin Mya Tun conceived and designed the experiments, performed the experiments, analyzed the data, prepared figures and/or tables, authored or reviewed drafts of the article, and approved the final draft.
- Puangwan Lapthanasupkul conceived and designed the experiments, performed the experiments, analyzed the data, prepared figures and/or tables, authored or reviewed drafts of the article, and approved the final draft.
- Anak Iamaroon performed the experiments, prepared figures and/or tables, and approved the final draft.
- Wacharaporn Thosaporn performed the experiments, prepared figures and/or tables, and approved the final draft.
- Poramaporn Klanrit performed the experiments, prepared figures and/or tables, and approved the final draft.
- Sompid Kintarak performed the experiments, prepared figures and/or tables, and approved the final draft.
- Siwaporn Thanasan performed the experiments, prepared figures and/or tables, and approved the final draft.
- Natchalee Srimaneekarn analyzed the data, prepared figures and/or tables, and approved the final draft.
- Nakarin Kitkumthorn conceived and designed the experiments, performed the experiments, analyzed the data, prepared figures and/or tables, authored or reviewed drafts of the article, and approved the final draft.

## Ethics

The following information was supplied relating to ethical approvals (i.e., approving body and any reference numbers):

The research was approved by the Institutional Review Board of Faculty of Dentistry/Faculty of Pharmacy, Mahidol University (COA.No.MU-DT/PY-IRB 2020/032.0906), Human Experimentation Committee of Faculty of Dentistry, Chiang Mai University (No. 45/2020), Human Research Ethics Committee of Faculty of Dentistry, Prince of Songkla University (EC6308-032) and Ethics Committee of Human Research Panel 2 of Faculty of Medicine, Khon Kaen University.

## DNA Deposition

The following information was supplied regarding the deposition of DNA sequences:

The sequences are available at GenBank: PRJNA1200226.

## Data Availability

The raw measurements are available in the Supplementary Files.

## Supplemental Information

Supplemental information for this article can be found online at http://dx.doi.org/10.7717/peerj.19137#supplemental-information.

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
