# Peer review of "A multi-center cross-sectional investigation of BRAF V600E mutation in Ameloblastoma"

_PeerJ, doi:10.7717/peerj.19137_

## Round 0.1 · original submission · Major Revisions

Please respond all the reviewers' comments.

Reviewer 1 ·

Basic reporting

No comment

Experimental design

No comment

Validity of the findings

No comment

Additional comments

Although this is a well-written and well-conducted study of relevance, minor revisions are needed to improve clarity and accuracy in certain areas. The authors will find detailed suggestions listed below, organized by manuscript section.
Introduction
• Line 84: Are the authors referring to BRAF protein or BRAF gene in the sentence: “V-RAF murine sarcoma viral oncogene homolog B1 (BRAF) is a serine/threonine kinase frequently affected by a somatic point mutation of BRAF V600E in human cancers”? The current approved name for the BRAF gene is “B-Raf proto-oncogene, serine/threonine kinase”, which the authors can verify using this link: https://www.genenames.org/. For the protein, the correct name is “Serine/threonine-protein kinase B-raf” (https://www.uniprot.org/uniprotkb/P15056/entry). Please, replace “V-RAF murine sarcoma viral oncogene homolog B1” with the correct name.

• Ensure that all gene symbols are italicized throughout the text. Human gene symbols should be written in uppercase letters and italicized, while protein symbols should be in uppercase letters and not italicized. For example: BRAF (human BRAF gene) and BRAF (human BRAF protein).
Material & Methods
• Line 140: the following sentence appears incomplete: “Interpretation of BRAF V600E mutation in the VE1 141 staining”.
• Line 169: Could you please confirm if the reverse primer sequence is correct? It appears to be mismatched with the BRAF sequence.
Results
• Lines 187-188: Could you please verify if the following sentence or the information in Table 1 is correct? “Furthermore, 40% of the AM cases were histologically diagnosed as follicular, followed by plexiform (32%) and mixed (28%) types”. Table 1 shows 28% for plexiform and 32% for mixed. Which value is correct?
• Consider placing supplementary table 9 in the main text as table 5.
Discussion
• Since 'MAPK' has already been defined as 'mitogen-activated protein kinase' upon its first mention in the Introduction, there is no need to redefine it in the Discussion.
Figures
• Consider removing the red circle and instead use an underline to highlight the codon. Also, ensure that the legend is updated to reflect this change, replacing 'red circle' accordingly.

Annotated reviews are not available for download in order to protect the identity of reviewers who chose to remain anonymous.

Reviewer 2 ·

Basic reporting

It is recommended to italicize gene names (e.g., BRAF).

Experimental design

The method for histologic subtyping of ameloblastoma (follicular, plexiform, mixed) was not described in the manuscript, which further undermines the significance of the relevant correlation analysis.

Validity of the findings

The power of statistical analysis generally increases by sample size. The authors emphasized significances observed in single institutions, such as the association between location and the mutation in cases from Mahidol despite no significance found in the total cases. Possible differences in demographics between the institutions have been suggested by the authors, which should be explained in detail to justify their assumption.

·

Basic reporting

Line 80 "the recurrence rates of both AM and OKC are reported to be high" This is subjective - it is better to report a range of recurrence rates as some may argue that these rates are rather low

Experimental design

Low sample size of OKC – given no association between BRAF and OKC, I was curious why it was necessary to include OKC in this study? I would consider removing OKC entirely from the study and potentially consider adding it as a part of a separate study.

Validity of the findings

You mention the use of adjunctive anti-BRAF targeted therapy to improve treatment outcomes, yet you also state no association between BRAF mutations and recurrence rate. Given that the gold standard treatment of ameloblastoma is surgical resection, what type of outcome improvements (if not recurrence rate reduction) are you hoping to achieve with the use of adjunctive anti-BRAF targeted therapy?

---

## Round 0.2 · Minor Revisions

I kindly request that you address the specific comments provided by Reviewer #2

Reviewer 1 ·

Basic reporting

No comment

Experimental design

No comment

Validity of the findings

No comment

Additional comments

The authors have addressed my suggestions satisfactorily. My only additional comment is to ensure that any references related to OKC removed from the text have also been deleted from the reference list.

Reviewer 2 ·

Basic reporting

Many "BRAF"s are still not being italicized. "BRAF" in "BRAF V600E" is also a gene.

Experimental design

Although the authors added the WHO classification as a reference, the criteria for the mixed subtype are not stated in the current WHO classification. The authors should provide the criteria for the mixed subtype since they try to report the correlation between BRAF V600E mutations and histological subtypes.

Validity of the findings

The authors should summarize the fact that "there was no significant association between BRAF V600E mutation and the location of ameloblastoma in the overall prevalence of our multi-center study" in the Abstract.

·

Basic reporting

no comment

Experimental design

Good move removing OKC data as it did not add much in this case. OKC data may be a separate study in the future with more cases

Validity of the findings

"BRAF inhibitors represent a potential alternative for patients contraindicated for surgical intervention, particularly in the geriatric population with extensive lesions". - old age not an absolute contraindication for surgery. Consider looking into BRAF therapy as an adjunct (not an alternative) to surgery to somehow optimize surgical outcome

---

## Round 0.3 · accepted · Accept

Authors have addressed all of the reviewers' comments.